# zIncubascope: Long-term quantitative imaging of multi-cellular assemblies inside an incubator

Anirban Jana[1,2,3], Naveen Mekhileri[1,2], Adeline Boyreau[1,2], Aymerick Bazin[1,2], Nadège Pujol[3], Kevin Alessandri[3], Gaëlle Recher[1,2], Pierre Nassoy[1,2], Amaury Badon[1,2] *

**1** LP2N, Laboratoire Photonique Numérique et Nanosciences, University Bordeaux, Talence, France, **2** Institut d' Optique Graduate School & CNRS UMR 5298, Talence, France, **3** Treefrog Therapeutics, Pessac, France

* amaury.badon@institutoptique.fr

**Data Availability Statement:** The data are available from the Open Science Framework database (link: https://osf.io/q6npr/).

## Abstract

Recent advances in bioengineering have made it possible to develop increasingly complex biological systems to recapitulate organ functions as closely as possible *in vitro*. Monitoring the assembly and growth of multi-cellular aggregates, micro-tissues or organoids and extracting quantitative information is a crucial but challenging task required to decipher the underlying morphogenetic mechanisms. We present here an imaging platform designed to be accommodated inside an incubator which provides high-throughput monitoring of cell assemblies over days and weeks. We exemplify the capabilities of our system by investigating human induced pluripotent stem cells (hiPSCs) enclosed in spherical capsules, hiPSCs in tubular capsules and yeast cells in spherical capsules. Combined with a customized pipeline of image analysis, our solution provides insight into the impact of confinement on the morphogenesis of these self-organized systems.

## Introduction

Since the advent of novel microfabrication and microfluidics techniques allowing a controlled production of three-dimensional cell cultures such as tumour spheroids or physio-mimetic organoids [1, 2], new needs in terms of optical imaging emerged. The search for novel techniques to improve the spatial resolution beyond the diffraction limit, with the constraint that acquisition times are inevitably restricted to a few seconds or minutes, is getting replaced with the quest for strategies that allow to monitor the growth and fate of these larger 3D supra-cellular assemblies. In particular, as mammalian cell cycle is of the order of one day, long-term imaging over several days to weeks is required to provide a dynamic perspective on biological processes. This is crucial for understanding phenomena with variable time scales, such as self-organization of cells, growth, and shape dynamics.

Current efforts are mostly aimed at tackling two issues. The first one consists in performing a full 3D reconstruction of the organoids/spheroids to unveil the inner structure. This may

**Funding:** This project was supported by grants from the French National Agency for Research (ANR-22-CE42-0019, ANR-21-CE18-0038, ANR-21-CE19-0029) and the Institut National du Cancer (PLBIO 20-135). The authors also acknowledge the financial support from the Grand Research Program LIGHT Idex University of Bordeaux, the Graduate program EUR Light S & T PIA3 ANR-17-EURE-0027 and GdR ImaBio. The funders had no role in study design, data collection and analysis, decision to publish, or preparation of the manuscript.

**Competing interests:** The authors have declared that no competing interests exist.

require to "clear" the sample to reduce light absorption and scattering [3, 4] with chemical treatment leading to sample fixation. Alternatively, optical techniques such as Optical Coherence Tomography (OCT) [5, 6] intrinsically improve the depth of light penetration and are exquisitely suited for live imaging since they are label-free methods and thus induce very low photo-toxicity. The second issue, which becomes crucial for screening applications, eg. in the pharmaceutical field, is to obtain statistics on the long-term fate of a large number of samples at the expense of information on the 3D structure. To this purpose, combining a sterile, temperature- and atmosphere-controlled environment in a sustained manner with optimal imaging capabilities becomes the main practical challenge. Conventional incubators that can precisely uphold such environments are available in all biology laboratories. However, displacing samples to and from the incubator to the microscopes is hazardous in terms of contamination risk and variations of the controlled environment. Automated imaging over time without repetitive manual intervention is thus highly sought-after.

For imaging over days, the standard method is to enclose the entire microscope in an environmental box, which takes up space in the laboratory and is costly [7–9]. Moreover, the microscope becomes unavailable over the imaging duration, hindering other experiments. Another readily practiced method is to mount an incubator chamber on the imaging platform of the microscope [7–9]. However, due to fluctuating conditions outside the chamber, condensation problems and light path disturbances may occur, compromising imaging for several days. Instead, installing an imaging setup in an incubator is easier and ensures a homogeneous environment for all the optical components, eliminating risks of differences in temperature/humidity.

Academic researchers and microscopy companies developed compact solutions that comprises both the microscope module and the controlled environmental conditions [10–12] or systems that fit directly inside a conventional incubator [13–16]. While the majority of commercial solutions offer high optical performances and automatic capabilities, the cost often remains prohibitive if needed outside dedicated imaging facilities. By contrast, academic solutions are usually cost-effective but specific to a targeted application. In this context, we recently developed a custom-made compact optical system dedicated to the observation of a large population of multi-cellular spheroids or organoids [17]. More specifically, given the method we used to produce these three-dimensional aggregates that consists in encapsulating cell in hollow hydrogel (namely alginate) spherical capsules [18, 19], the samples are particularly sensitive to stage motion. To fulfil this requirement without compromising statistics, our "Incubascope" was designed to fit in a tabletop incubator and offer imaging over a large field of view in a single frame, i.e. without any sample motion [17]. In this first version, the apparatus was designed for imaging around 100 aggregates with a diameter of about 300 μm but at moderate spatial transverse and axial resolution (3 μm and 50 μm respectively). While this apparatus was particularly suitable to estimate the overall fate of a population of multi-cellular aggregates, the ability to investigate more finely at different spatial scales biological phenomena involving living spheroids or organoids was hindered due to the poor resolution that was achieved.

In this work, we present an augmented version of our approach, which we coined the zIncubascope. The imaging setup has been improved from a hardware point of view in terms of compactness, versatility and the capability to image in the z-axis of the referential plane. It has also been supplemented by a data analysis pipeline. Indeed, recognizing the importance of space inside an incubator, our zIncubascope has been engineered to occupy significantly less room compared to its predecessor. This gain in compactness is accompanied by a decrease in the number of optical elements which in turn provides a simpler and more robust platform. Additionally, the system is now compatible with microscope objectives (MO) of different

magnifications, so that spatial resolution and field of view can be adapted to the sample and the purpose of the experiment. Finally, we developed custom-made codes to control the apparatus and to perform quantitative data analysis.

In the following, we first describe the technical details and design principles that underpin our imaging platform, to enable readers to grasp its features and capabilities, and to facilitate replication of the platform. Then, we provide a series of examples, highlighting the performance and versatility of the system across different biological samples, which are human induced pluripotent stem cells (hiPSCs) enclosed in spherical capsules, hiPSCs in tubular capsules and yeast cells in spherical capsules. In particular, we present evidence that the zIncubascope allows us to gain insight in the impact of confinement on the morphogenesis of these self-organized systems.

## Imaging setup

In this section, we describe the design and performances of the zIncubascope. More specifically, we detail (1) all opto-mechanical components required to achieve its assembly and the optical performances obtained, (2) the focus mechanism operation and capabilities and (3) the electric components.

### Imaging opto-mechanical components

In this new version of a microscope inside an incubator, we have developed an imaging system with a minimal number of opto-mechanical components (Thorlabs) to ensure simplicity and ergonomics. For sample illumination, a single LED (M625L4 LED, Fig 1) is used in combination with an aspheric lens (L1, Fig 1) to generate a collimated beam. A right-angle mirror mount (KCB1/M, RM1, Fig 1) is used to redirect the beam from the horizontal to the vertical plane, illuminating the sample from above. The transmitted light is collected with an objective, whose magnification can be varied between 4X and 20X (Olympus Plan Achromat 4X/10X/

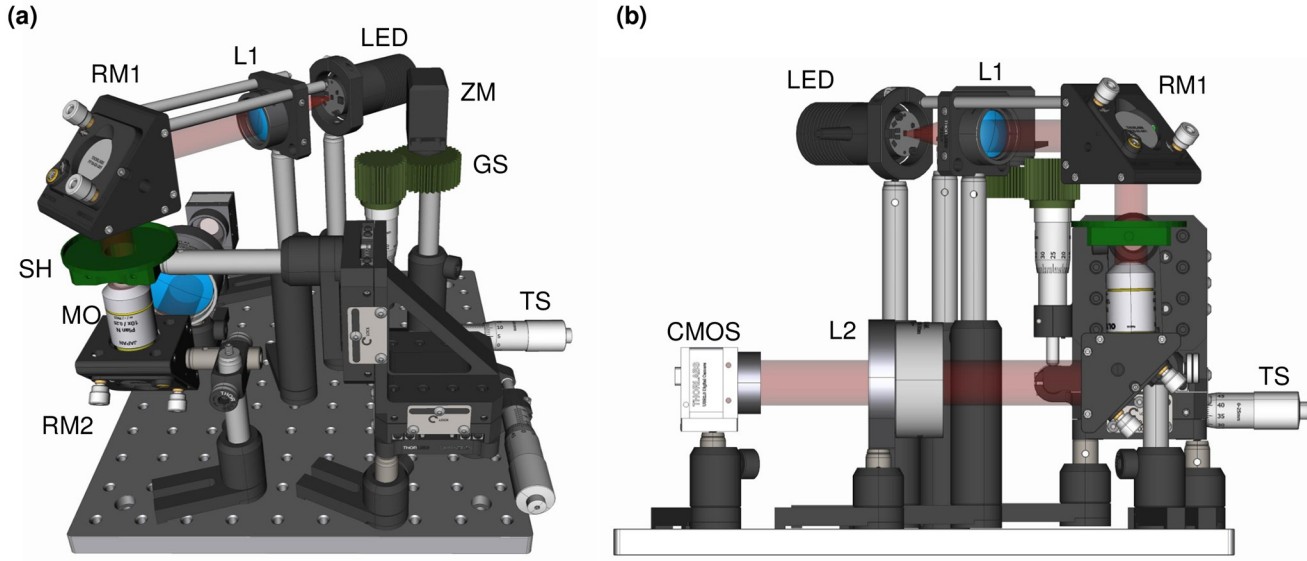

**Fig 1. 3D rendering of the optical setup.** (A) An orthogonal CAD representation of the zIncubascope, with illumination shown as a red cylinder, emerging from L1. (B) A side view of the setup to highlight the optical path and the CMOS.

20X, MO, Fig 1) depending on the specific needs described in the results section. This plan-achromatic class of objectives provides relatively high numerical apertures (NA = 0.10–0.4) for visible wavelengths, with large working distances (1–20 mm). This allows samples to be imaged through standard plastic petri dishes (bottom and slide thickness 1 mm, diameter = 35 mm), convenient for placement and media exchange. Customized 3D resin-printed platforms (sample holder SH, Fig 1) are used to securely mount these petri dishes. The dish fits snug in this mount to avoid displacements arising from external vibrations or during change of media in the dish. The petri dish holder is attached with a post onto a XYZ-translation stage (LX30/M, TS, Fig 1). This translation stage can cover a volume of 25x25x25 mm$^3$ which enables easy 3D localization of the region of interests of the sample(s) in the petri dish. The objective is fixed on a right-angle mirror mount (KCB1/M, RM2, Fig 1) that reflects the signal to the horizontal plane onto a tube lens (AC508–150-A-ML, L2, Fig 1), that ultimately focuses it onto a CMOS camera. The mirror mounts above and below the sample are easily screw-adjustable, facilitating rectifications in minor drifts of optical alignment that could appear with time and usage. These corrections are necessary to obtain images with homogeneous contrast and minimal aberrations. Utilization of three-dimensional space reduces the overall footprint of the system, without adding major complexities in the installation of the components. This system is designed to be modular, permitting reconfiguration or further compaction for specific purposes. It is even possible to accommodate an additional zIncubascope in the same incubator providing its size be at least 118 liters, which corresponds to the typical dimensions of incubators found in culture rooms. The CMOS camera used in the setup is a 20-megapixel sensor with a pixel size of 2.4 x 2.4 μm$^2$ (Basler acA5472–17um). As seen in Fig 2A, in the imaging configurations of interest, the field of view (FOV) can be set at 6 x 4 mm$^2$ (with the 4X objective), 2.3 x 1.6 mm$^2$ (10X) and 1.15 x 0.8 mm$^2$ (20X). The theoretical resolution achieved is 3.8 μm (4X), 1.52 μm (10X) and 0.95 μm (20X). In all cases the detection is diffraction limited since each pixel on the camera is equivalent to 1.08 μm (4X), 0.432 μm (10X) and 0.216 μm (20X). This is in agreement with the smallest elements visible on a USAF 1951 resolution target in the 3 different optical configurations (Fig 2B and 2C).

## Z-Drive system

Organoids or 3D yeast assemblies imaged in this work have typical sizes in the range of 100–500 μm, but they are not monodisperse. Consequently, it is unlikely that all samples in the FOV have the same focal plane, thus leading to the acquisition of blurred images for most of them. Similarly, even if the focal plane is initially chosen to have most samples in focus, the growth of the aggregates will inevitably alter the focus. This is all the more critical when objectives have high NA. Therefore, to acquire data from the full FOV, imaging in a single plane at any given time step is insufficient. Hence a simple servo motor (ZM, Fig 3A) (Ref: MG996R, TiankongRC), with a suitable gear-train (GS, Fig 3A) is mounted on the Z-micrometer of the translation stage to ensure that in-focus images of all samples are taken at each time point.

The resolution of the micrometer (LX30/M) is 10 μm in displacement, and since we are not aiming for sub-cellular imaging (typical size of cell 10 μm), the smallest stack size we chose is 10 μm. We observed that the servo motor used in the setup practically tends to lock and lacks consistency in angular rotations below 5 degrees. To address this limitation, we have set the minimum rotation step at 5 degrees. Considering these constraints and the fact that the servo motor has a maximum rotational range of 180 degrees, we have opted for a gear ratio of 36:25 (driver: driven) with the theoretical goal of achieving a minimum stack size of 10 μm over a range of 360 μm (Fig 3A). The gears are printed in resin (envisionTEC HTM140 V2) using an Envision D4K 3D-printer. The gear (G1, in Fig 3A) is secured with epoxy on a rotational stage

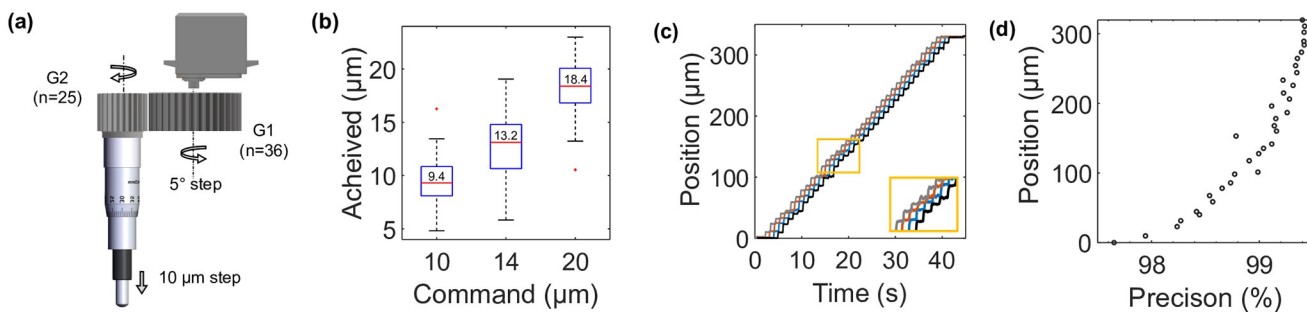

**Fig 2. Optical performances of the experimental setup.** Field of view and resolution obtained with (A) 4X (red), scale bar = 500 μm, (B) 10X (gold), (C) 20X (blue) objectives.

with a sleeve that is mounted on the shaft of the motor. The driven gear (G2, Fig 3) is printed in the shape of a cap that is pressed fit on the knurled thimble of the micrometer. The motor-gear setup is mounted on a post and can be isolated from the micrometer to adjust the stage manually if needed.

**Fig 3. Mechanical performances of the axial scanning module.** (A) Schematic of the gear-setup and motor mounted on the Z- micrometer. (B) Step-sizes achieved for step-size commands. (C) Compilation of the steps achieved by the stage over 4 different runs at 10 μm step size commands with 1 second of resting time between successive steps. The horizontal shift in the curves is for easy visual interpretation and has no physical significance. In snippet, a magnified section of the compilation. (D) Variation of the precision obtained from the 4 different runs (shown in C) at different positions.

Through repeated testing using a customized setup (see S1 File, section 3), the accuracy of the system has been evaluated for different step sizes with different pause intervals between each step to verify delays in movements. Overall, we have observed that the stage movement exhibits an accuracy ranging between 80–90%. However, local precision is very high, around 99% for all step sizes, irrespective of pause intervals. In the applications described below, 3D reconstruction is not a primary aim compared to capturing events at different focal planes, and hence high precision time-lapses at selected focal planes without significant depth displacements are sufficient.

### Electrical and control components

For illumination, the LED is controlled by a LED-driver (LEDD1B) modulated by an input voltage signal, generated through an Arduino (Arduino DUE). The Arduino is also used to control the Z-motor, but the motor is powered and grounded with an isolated 5V USB input to avoid attenuation in the modulation of the illumination. The Arduino is commanded using custom Python codes (see S1 File, section 4) with a universal graphical user interface to ensure user-friendly control of the imaging system.

## Results

To validate the design, modularity, and performances of the zIncubascope, we have explored a variety of biological samples with different system configurations. In this section, we will first monitor the luminogenesis and growth of hiPSCs in spherical capsules followed by the study of their morphology in tubular capsules at high spatial resolution. Finally, a demonstration of the versatility of our approach will be demonstrated by imaging and analysing the growth of a large population of spherical capsules containing yeast at a reduced spatial resolution.

Using the zIncubascope, we have studied the growth and behaviour of 3D colonies of hiPSCs in alginate spherical capsules. These capsules are obtained using a microfluidic encapsulation technique called the Cellular Capsule technology [18]. Technical details are provided in S1 File. The capsules typically have an outer diameter and a core diameter of 450 μm and 300 μm, respectively, with variations of about 10% from one capsule to another [18]. Practically, from a given encapsulation batch, a few capsules are pipetted and seeded at the bottom of a poly-lysine coated petri dish. In capsules containing a prototypical extracellular matrix, namely Matrigel (Corning: 354234), hiPSCs self-assemble and form cysts, i.e. spherical closed epithelia [20]. Depending on the initial seeding cell density, only one or several cysts per capsule are nucleated and grow in the capsule compartment. These cysts have different sizes and are located at different depths inside the capsules. We thus took advantage of the z-drive system that we implemented to analyse the morphometric parameters of the cysts in time.

### Growth of 3D hiPSCs colonies in spherical alginate capsules

Fig 4A shows a full field view ($\sim$ 2.3 x 1.6 mm$^2$) of cysts-containing capsules in a petri dish. Imaging is performed with a 10X objective for 100 hours (Day 1 to Day 6) from 24 hours after encapsulation (Day 0). Image acquisition is performed every 90 minutes with z-step set at 40 μm over a range of 80 μm from Day 1 to Day 3 since the dynamics are slow and the cellular assemblies are relatively small (S1 Video). From Day 4 till Day 6 images are acquired every 60 minutes over a stack of 240 μm with a step-size of 80 μm (S2 and S3 Videos). Culture medium is replenished on Day 1, Day 2, and Day 4 without removing the petri dish from the imaging stage. In this configuration, at least 10 capsules can be imaged in full FOV with sufficient resolution for analysis and post-processing. As seen in Fig 4B, some cysts are out of focus for a

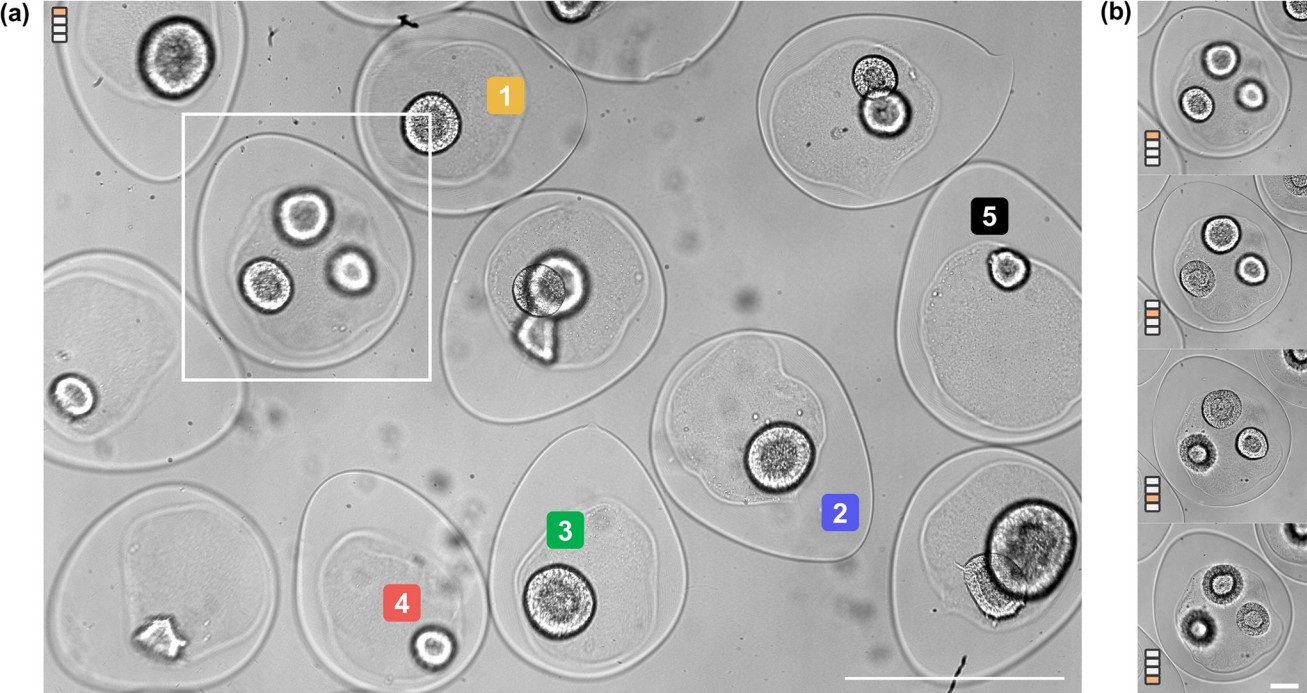

**Fig 4. 3D observation of hiPSCs in spherical capsules.** (A) Full-field view of the petri dish (2.3 x 1.6 mm$^2$) seeded with capsules with the stage positioned at 80 μm above an initial reference point (position = 0 μm) at t = 126 hours. Scale bar = 500 μm. (B) Cropped image of the capsule marked in (A) at four different focal positions (0, 80, 160, 240 μm), allowing visualizations of the three cysts at different depths at t = 126 hours. Scale bar = 200 μm.

given z position of the stage. However, with the movable z-stage, all cysts in the field of view can ultimately be brought in focus (Fig 4B), and thus be easily captured and analysed.

Within the first 2 days after encapsulation, hiPSCs-laden capsules reside in a medium containing the ROCK inhibitor Y27632, which is known to improve the survival of hiPSCs upon dissociation from 2D culture [21]. Y27632 is then removed on Day 2 to avoid alteration of the metabolism of hiPSCs [22] and we observed that the cells cluster (Fig 5A, 59h post-encapsulation). A hollow cyst-like structure with a lumen appears at the end of Day 2 (Fig 5A, 62–65h), and the cell-layer continues to grow with expansion of the lumen (Fig 5B) over the following days. Quantification of the growth rate of these lumenized structures is performed from the end of Day 3 to Day 5. Using a customized analysis (see S1 File, section 2.1), the thickness of the closed epithelium, *t*, and its internal radius, *r*, are measured at 70 positions around the cyst. (Fig 5C). The cellular volume is calculated using the spatially averaged thickness and lumen radius, considering the system as concentric spheres ($4/3 \times \pi \times ((r+t)^3 - r^3)$). Fitting of the cellular volume with an exponential yields the apparent population doubling time (PDT) [20] assuming the volume increment is governed by cell proliferation and that changes in individual cell volumes can be neglected. The PDT obtained using an exponential fit on the data is 15 ±0.39 h (N = 5) (Fig 5D). This value is in agreement with proliferation rates reported in the literature for embryonic and induced pluripotent stem cells [23–25], suggesting that death rate is negligible in our cellular capsules. Our analysis also reveals that the thickness of the cell layer increases linearly with the lumen radius. Beyond a critical lumen radius of around 40 μm, there is a change in the slope, and more specifically an enhanced thickening with lumen radius (Fig 5E). This effect can be described using a generic morpho-elastic model by assuming that cyst growth is anisotropic [26]. However, deeper biological investigation, which is beyond the

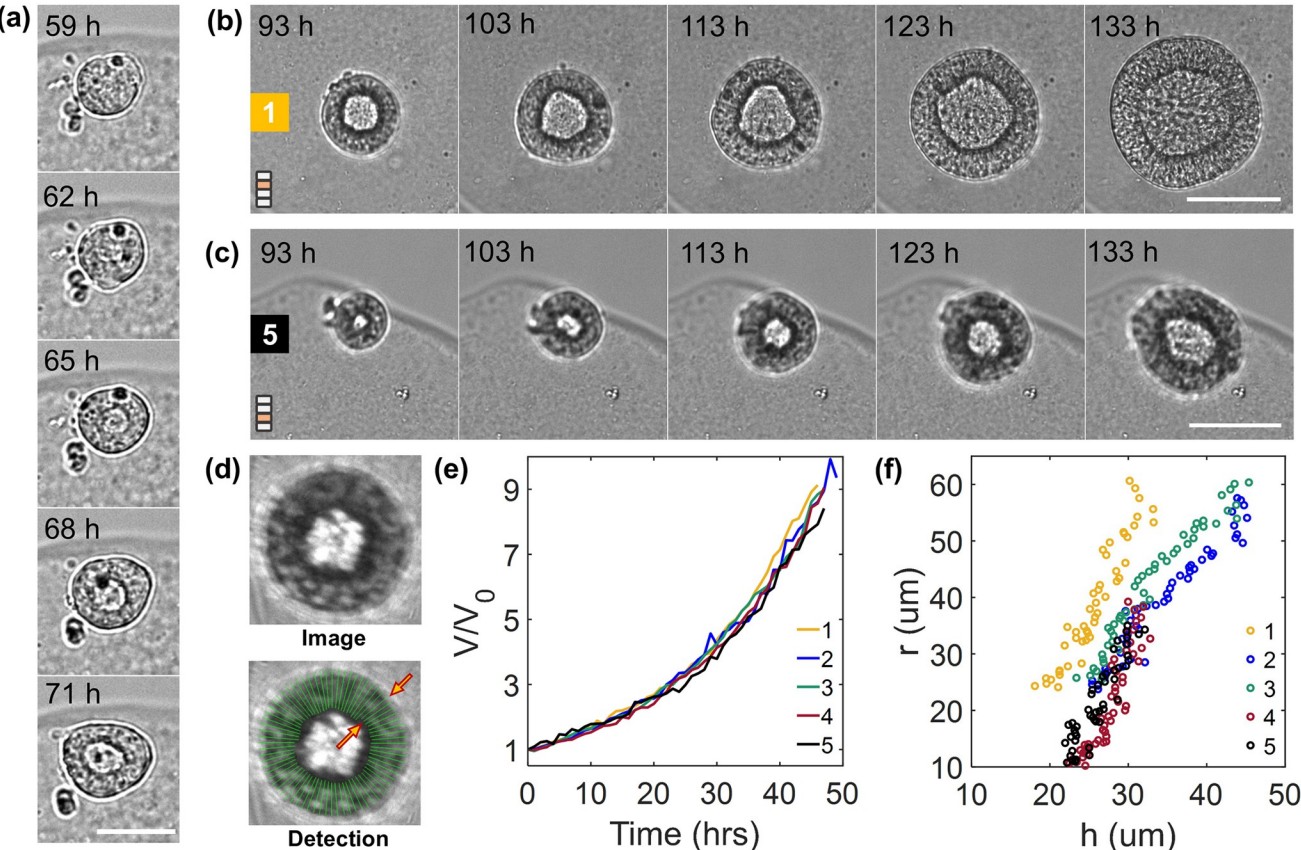

**Fig 5. Spatio-temporal analysis of cyst shape in spherical capsules.** (A) Emergence of a central lumen from a cluster of cells. Scale bar = 50 μm. (B) Time lapse of cyst growing in capsule 1 and capsule 5 marked in Fig 3A over 40 hours at different preset z-positions over time. Scale bar = 200 μm. (C) Detection of the cellular layer of the cyst to estimate the lumen radius and the thickness of the cyst. (D) Variation of the volume (normalized with the initial volume of each cyst) of the cell layer of the cysts (marked in Fig 4A) with time. (t = 0) refers to 93h post encapsulation. (E) Evolution of the lumen radius of the cysts with the thickness for different cysts (marked in Fig 3).

scope of the present paper, would be required to decipher the underlying mechanism. Note that in all experiments reported here, cysts have not reached confluence, i.e. hit the walls of the capsule and get confined, at Day 5. As reported in [20], this phenomenon usually occurs after Day 6 and is followed by a collapse of the lumen due to progressive thickening of the cellular envelope.

## hiPSCs growth and morphology in tubular alginate capsules

In a D'Arcy Thomson perspective [27], the second case of study described here consists in investigating how 3D hiPSCs colonies that spontaneously self-assemble into spherical cysts as shown above, will accommodate in a lower symmetry capsule. To do that, we encapsulated hiPSCs in hollow alginate tubes. The fabricated tubes (see Supp. Materials, section I C) typically have an outer diameter and a core diameter of around 260 μm and 170–185 μm, respectively, and has a length of around 1 m upon formation. In contrast to the closed spherical capsules, cysts are never completely confined in tubes. There is a partial release of confinement along the main axis of the tube. We have studied the growth dynamics and the role of partial confinement on the morphogenesis of these tubular cellular systems. A large FOV coupled

with high spatial resolution and z-imaging capability offered by the zIncubasscope has been utilized to acquire and analyse these behaviours.

As for capsules, the 10X configuration has been used. In the present imaging configuration, a length of around 1.6 mm of the tube can be accommodated in the FOV of 2.3 x 1.6 mm$^2$ (Fig 6A). Practically, on Day 1 (24h after encapsulation), a section of the tube is selected, snipped, and placed on a poly-lysin coated petri dish. In the FOV of 2.3 x 1.6 mm$^2$ shown in Fig 6A, we observed (S4 Video) that cells are initially isolated or form tiny clusters of less than around 10 cells over the first 20 hours following encapsulation (Day 0). The cell density along the tube is heterogeneous and hence multiple cysts are formed. These assemblies seem to explore their surroundings along the length and width of the tube. With multi-focal acquisitions performed with steps of 20 μm, cells at different azimuthal positions can be resolved and monitored as they grow (Fig 6B).

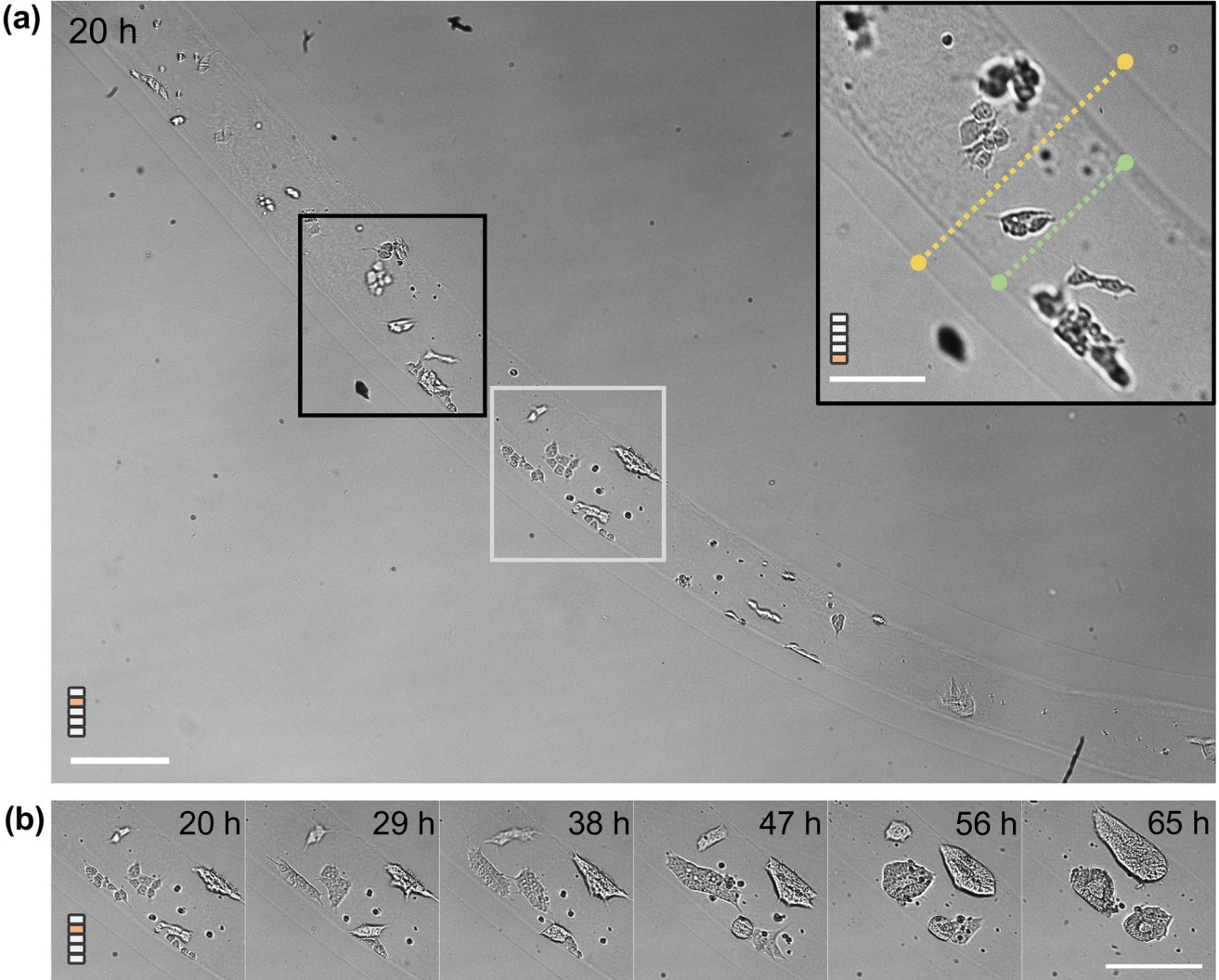

**Fig 6. Early-time observation of hiPSCs growth in tubular capsules.** (A) Full-field view of the petri-dish with an alginate tube at a stage positioned 30 μm above an initial reference position (0 μm) at 20 hours after encapsulation. Scale bar = 200 μm. Insert: A region of interest in the tube at a focal plane 80 μm above the reference position. The inner and outer diameter of the hollow tube is shown with the green and yellow dotted lines, respectively. Scale bar (for insert) = 100 μm (B) In the region marked by a white ROI in (A), appearance of clusters from single cells is shown over 45 hours (day 1 to day 2) at 30 μm above an initial reference depth from the beginning of imaging. Scale bar (for B) =200 μm.

As in the case of spherical capsules, nucleation of a central lumen in small aggregates is observed at the end of Day 2. The nascent cysts are observed to grow and glide along the axis of the tube, till they get confined radially by the alginate wall. From an initial spherical shape, the aspect ratio of the cysts increases i.e. they become cylindrical cysts (Fig 7A). Note that, since the time required for the cyst to be confined depends on the initial size of the cyst, and thus the initial cell density distribution, all events from cell clustering, lumen formation, spherical cyst growth to cyst deformation can be captured within a single experiment using our setup. We have analysed one single cyst as it grows in a confined configuration from the end of Day 3 to Day 4 (Fig 7A). Fig 7B shows the relative variations of the length and width of the cyst normalized by their dimension (length and thickness, respectively) $L_0$ taken at reference time (corresponding to 91.5 h post initiation of imaging) as a function of time. An increase of 90% in the axial length and 10% increment in the radial direction within 18 hours of growth are observed. The centroid of the cyst undergoes minimal displacements with relatively higher displacements in the unconfined axial direction (Fig 7B, insert). This implies similar proliferating fronts on both sides of the cyst axis and confinement of the system radially.

Growth of the tubular cyst is characterized by measuring the volume of the cellular layer according to $V = A \times D$, where $A$ is the projected area and $D$ is the diameter (shown as width in Fig 7B) of the tubular cyst. Similar to the spherical cysts described above, the system undergoes an exponential volume increase with an apparent PDT of 15 hours (Fig 7C), suggesting that partial radial confinement does not slow down the proliferation rate of the cells within this cellular assembly. However, striking changes in the local morphology of the cellular envelope are frequently observed (Fig 7D). The tip of the tubular cyst, which has no interaction with the alginate tube exhibits a cell layer that becomes progressively thinner and flattens as it continues to expand in the tube. In contrast, the cell layer in contact with the inner alginate wall thickens with time. Fig 7E displays the axial and radial thicknesses of the epithelium normalized to the local thickness at reference time t = 0. The radial thickening is concomitant with the thinning of the cell layer at the tip. In Fig 7D, at time t = 111 h, or equivalently t = 19.5 h post-reference time, a cyst, resulting from the "docking" between two cysts (on the left), comes in contact with another cylindrical cyst (on the right). No striking change in the evolution of the thickness of the cyst tips is observed, suggesting that the interaction between cysts does not affect the overall growth dynamics of the cellular assemblies of hiPSCs.

Using the full FOV provided by the zIncubascope with the 10X objective, we then sought to inspect the processes that occur when multiple cylindrical cysts encounter each other. We most often observe that a layer of cells remains visible between the colliding cysts, indicating that, instead of coalescence, cysts only dock to each other and leave "scars". The two cell layers do not fuse but get distorted as the cellular assembly continues to grow (Fig 7F and 7G (top)), suggesting local differences in cell proliferation rates. Acquiring at multiple focal depths reveals peculiar features along the cellular tube axis at late stages of growth. In the reference focal plane (z = 0 μm), Fig 7G (bottom) highlights that the free end of the cyst persists to be thin and flattened. At a depth of 100 μm above the reference position, in the region of confrontation of the cysts, a cellular ridge spanning across the width of the tube is clearly observed and remains persistent for hours. These unique features can be easily captured and analysed over long periods of time, providing insights into the behaviour of partially confined hiPSCs assemblies.

## Yeast encapsulated in capsules

The experiments reported above have focused on the study of mammalian cells. To demonstrate the versatility of our approach, we then investigated yeast cells (*Saccharomyces*

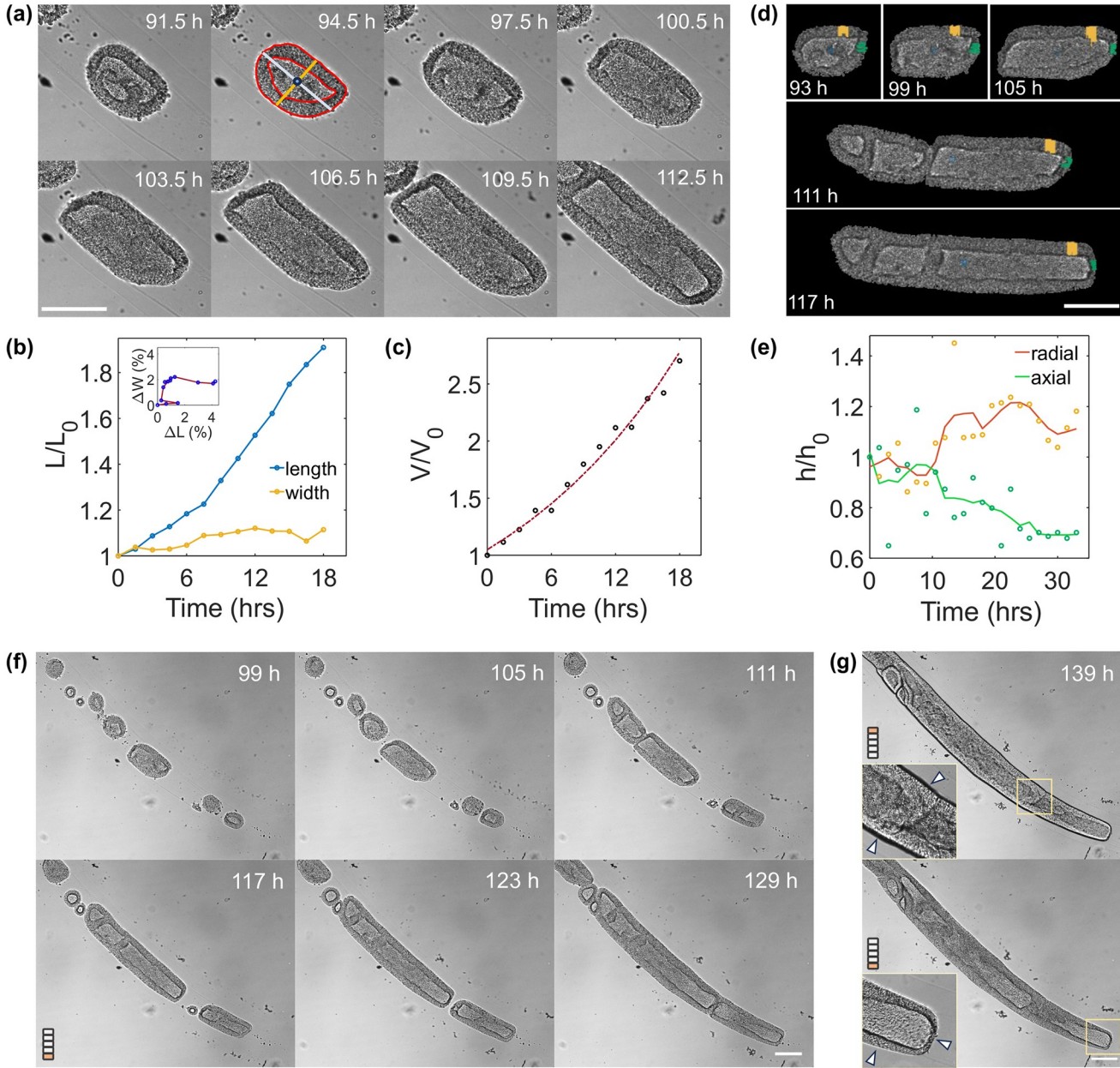

**Fig 7. Multi-scale spatio-temporal observation and analysis of hIPSCs in tubular capsules.** (A) Timelapse of an isolated cyst from 91.5 hours after encapsulation. Scale bar = 200 μm. (B) Variation of length (resp. width) of the cyst (as shown in (A)) over 18 hours with $L_0$ being the initial length (resp. width) at reference time t = 0 (taken 91.5h after encapsulation). Length is marked by the blue line, width by the gold line. (C) Volume variation (normalized with volume $V_0$ at t = 0) of the cell layer, as outlined in red in (A). (D) Post-processed time-lapse of the cyst assembly shown in (A) as it proliferates. The thickness of the cell layer at the radial (gold lines) and axial (green lines) positions of the cyst front are measured and shown in gold and green, respectively. Scale bar = 200 μm. (E) Variation of the average thicknesses measured in the positions shown in (D). The lines are visual guides. $h_0$ is the axial (resp. radial) thickness normalized by the value at time t = 0 corresponding to 91.5h post-encapsulation. (F) Full-field view time lapse of cysts as they proliferate in the tube captured at 100 μm above the reference position (0 μm). Scale bar = 200 μm. (G) Images of the tube at depths of 0 μm and 100 μm at t = 139h highlighting different features of the tube. (Top) The cell layer trapped in the lumen of the tube revealing incomplete fusions. (Bottom) The difference in cell-layer width at the axial tip and radial positions of the elongating cyst. Scale bar = 200 μm.

*cerevisiae*) encapsulated in alginate capsules. Yeast cells are generally between 5 and 10 μm in diameter and proliferate in the form of clusters in the capsules that can be easily monitored with a low resolution objective (Olympus Plan Achromat 4X, NA = 0.1). The capsules are seeded on a poly-Lysine-coated petri dish. Since the proliferation rate is expected to be much faster than mammalian cells (in the 1.5–4.5 h range in fermentative conditions depending on temperature, with an optimal temperature around 20˚C) [28, 29], a time step of 15 minutes is set, and a single depth of focus is imaged (S5 Video). The culture media is unchanged over the course of this acquisition. The imaging setup is placed in ambient temperature to match the sub-optimal fermentation growth conditions in wine or beer production and acquisition is initiated 1-hour post-encapsulation.

In this configuration, we obtain a FOV of 6 x 4 mm² that allows around 100 capsules to be imaged in a single image (Fig 8A). As yeast division time is short, we readily observe the

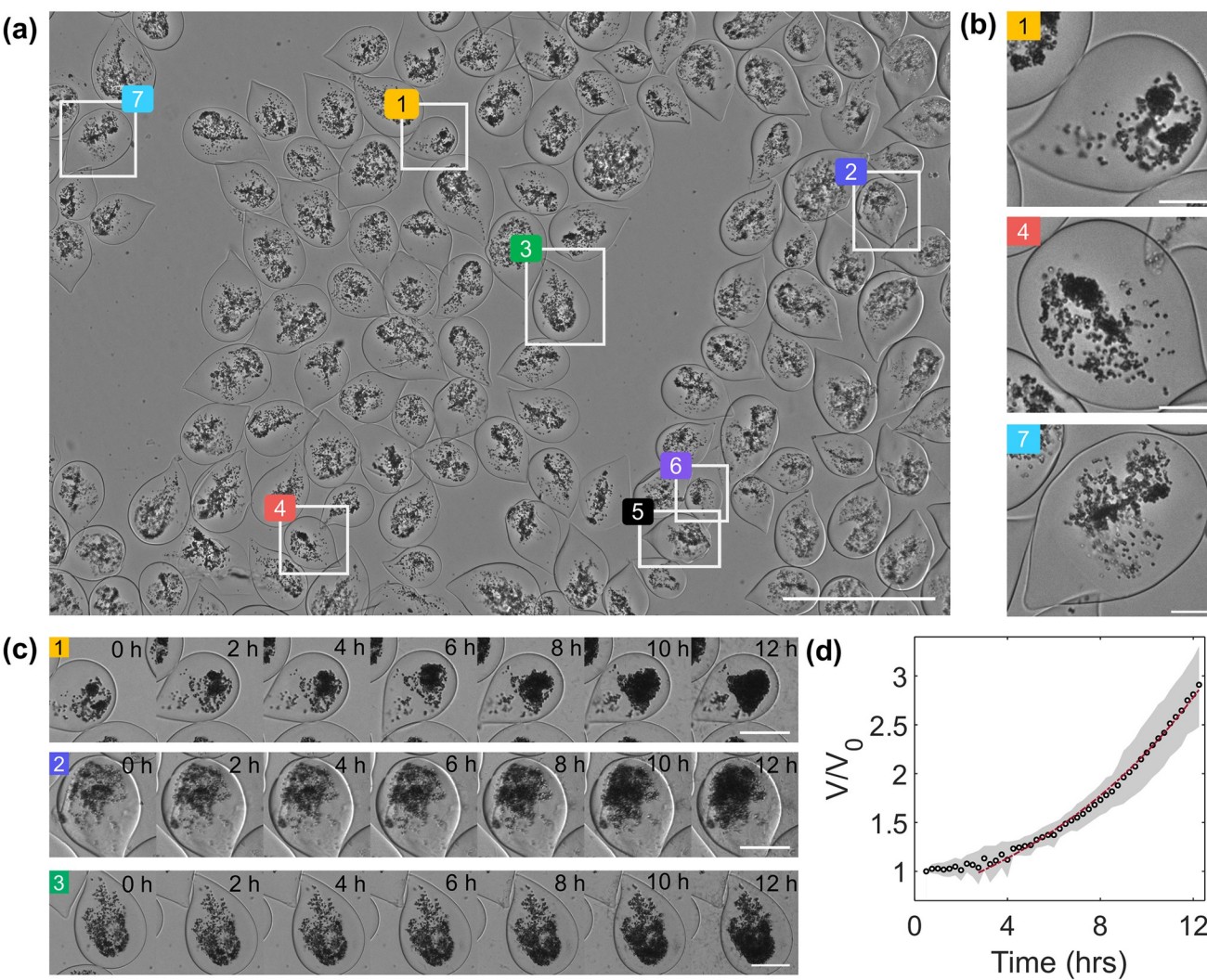

**Fig 8. Large-scale observation of encapsulated yeast.** (A) Full-field view (6 x 4 mm²) of the petri dish seeded with capsules encapsulated with yeast. Scale bar 1 mm. (B) Expanded version of capsules from the extremities of the FOV highlighting the homogeneity of the illumination and focal plane. Scale bars = 100 μm (C) Time lapse of proliferation of the yeast aggregates in capsules (marked in (A)) from 5 hours post-encapsulation over 12 hours. (D) Variation of the volume (normalized with the initial volume of each aggregate) of the yeast aggregates (marked in (A)) with time. (t = 0) refers to 5 h post encapsulation. An exponential fit is performed from 75 minutes.

expansion of yeast cells inside capsules. In only few hours, most of the capsules get dark as the aggregates grow and densify (Fig 8C). To quantitatively assess the proliferation rate of yeast cells, we take benefit of the fact that, even on such a large FOV, the homogeneity of the illumination is constant, which permits to perform intensity measurements (Fig 8B). As a proof-of-concept, we restrict our study to 6 capsules taken randomly at different spots of the FOV to estimate the volume of the yeast cluster as a function of time (S1 File, section 2.3). A characteristic depth or thickness of the aggregate is calculated for each pixel using the Beer-Lambert's law. The volume is derived by multiplying this length by the area of the pixel. The integration of these elementary "pixel volumes" yields the volume at a given time step. An exponential fit is then performed on the normalized volumetric growth from 4 hours after initiation of imaging to estimate the doubling time of the aggregate, which is found to be equal to $330 \pm 35$ minutes ($n = 6$), in agreement with the range of reported values in the literature [28, 29]. The delay in attaining an exponential regime of growth can be assigned to a lag phase in which yeast cells are metabolically active but still do not proliferate [30]. Besides the encapsulated yeast, we observe free floating yeast cells in the medium, which might be due to ill-formed capsules generating leaks. Growth of these free yeast leads to a rapid coverage of the petri dish, which darkens the field of view and imposes to stop acquisition after only 12 hours (S5 Video).

## Discussion and conclusion

In this work, we have developed the zIncubascope, an imaging platform which is a tailored solution for medium-to-high-throughput monitoring of cell assemblies inside an incubator. The utility and performances of our setup have been exemplified by imaging hiPSCs cysts encapsulated in alginate capsules of spherical and tubular shapes over days as well as encapsulated yeast clusters over hours. The spatial and temporal resolution achievable with this setup allow us to perform quantitative analysis on a statistically sufficient number of samples in a single experimental run. We developed a customized pipeline of image analysis codes which, when combined with our experimental setup, offers a compact, versatile, and cost-effective imaging solution.

First, our apparatus was designed to fit on a small breadboard (25x30 cm) in order to be accommodated inside an incubator. In this new configuration, two zIncubascopes can even be housed inside standard incubators, thus increasing the capability of imaging either more numerous, more diverse samples or different experimental conditions. The imaging setup has been designed with the perspective of easy installation and maintenance. This solution is based only on the shelf components and 3D printed parts for which CAD files are provided (see GitHub page [31]). The ergonomy of our solution also lies in the codes we have developed in Python to control the apparatus. Instructions on how to install and adapt these open-source codes are also provided.

The second main advantage of our experimental apparatus is its versatility in terms of field of view and spatial resolution. In contrast to our previous work [17] where the system was designed for a large parfocal length microscope objective (MO), here our system is compatible with all MO whose parfocal length is equal to 45 mm. By simply changing the MO, we can explore different systems at different spatial scales as illustrated in this work. It is worth pointing out that all experiments were carried out in bright field mode, as we decided to keep the hardware as simple as possible. Integrating an epifluorescence mode will indeed augment the capacities at the expense of increased complexity and price, which is here below 4k € (without the computer and incubator). If needed, our system can also be easily upgraded with a motorized stage to increase the FOV. The solution we provide for the motor driven z-stack

capabilities can be extended in the 3 dimensions. This approach is both ergonomic and cost-effective. Alternatively, compact commercial motorized stages that provide better performances are often compatible with a Python control but the price of such a system is usually more expensive than the total price of the zIncubascope.

Thirdly, we have addressed the imaging analysis difficulty by developing custom made codes both in Python and Matlab (available at [31]). These streamlined codes are convenient to use and to adapt to similar cases without advanced programming knowledge. No deep-learning or GPU use that can be more challenging to implement are necessary. Yet, our simple analysis enabled us to address diverse biological behaviours and unveil unreported behaviours of hiPSCs lumenized clusters. Indeed, while self-assembly of hiPSCs into cysts that resemble lumenized epiblast found in the development of an embryo [32] has been reported *in vitro* previously [20, 33–35], we bring evidence for new morphological observations. First, in contrast with other advanced methods consisting in tracking single cells with rainbow reporters [25], we could estimate the population doubling time (PDT) of hiPSCs in these 3D cysts by accurate monitoring of the volume of the epithelial layer. Second, we obtained hints that cyst growth seems to occur in two phases (Fig 5(e)). Third, the qualitative observation that i) growth of cylindrical cysts in conditions of partial confinement in a tubular capsule leads to the spontaneous emergence of anisotropic morphology marked by thin tips, and ii) cylindrical cysts docking most often leads to incomplete coalescence, suggesting that merging of hiPSC epithelia is inhibited via their basal side reinforced by a basal membrane. We hope that the dual advantage in the hardware implementation and the customized analytic pipeline will be a major asset for a wider dissemination.

Our solution was shown to be suitable and sufficient to obtain sharp images of hollow three-dimensional samples, without the use of deep-learning techniques [36]. However, scattering and aberrations tend to prevent observations of thick samples in depth with enough contrast such as in the case of solid spheroids. In such configurations, projected images have been used to monitor and analyse the growth of multicellular assemblies [18]. Finally, even with a motorized stage in the axial direction, it is of note that our apparatus does not perform optical sectioning, thus true 3D imaging. Using the framework of the zIncubascope, with the addition of minimal opto-mechanical components, we aim to implement advanced imaging techniques like optical coherence tomography or light field microscopy to address the current constraints.

## Supporting information

**S1 File. File that provides details on culture methods, image analysis, on the setup to measure the performances of the motor driven z-stack capabilities, the acquisition code and the experimental setup.**
(PDF)

**S2 File.**
(TEX)

**S1 Video. Timelapse of an hiPSCs cyst growing in a spherical capsule (cropped from the full FOV) from Day 1 to Day 3.**
(DOCX)

**S2 Video. Timelapse of hiPSCs cysts growing in spherical capsules from Day 4 to Day 6.**
(DOCX)

**S3 Video. Timelapse of 3 hiPSCs cysts growing in a spherical capsule imaged in different planes from Day 4 to Day 6.**
(DOCX)

**S4 Video. Timelapse of hiPSCs proliferating in a tube of alginate from Day 1 to Day 5.**
(DOCX)

**S5 Video. Timelapse of yeast proliferating in spherical capsules over 12 hours.** Scale bar = 500 μm.
(DOCX)

**S1 Fig.**
(TIF)

**S2 Fig.**
(TIF)

**S3 Fig.**
(TIF)

**S4 Fig.**
(TIF)

**S5 Fig.**
(TIF)

**S6 Fig.**
(TIF)

## Acknowledgments

We thank all the other members of the BiOf (Bioimaging and Optofluidic) team, in particular Camille Douillet, Aurélien Richard, Laetitia Andrique, Léon Rembotte, Fernanda Lopez-Garcia, for beta testing the device and suggesting improvements. We are also grateful to all the colleagues that encouraged us to develop a user-friendly version of the zIncubascope by expressing enthusiastically their need for such an instrument.

## Author Contributions

**Conceptualization:** Anirban Jana, Gaëlle Recher, Pierre Nassoy, Amaury Badon.

**Data curation:** Anirban Jana.

**Formal analysis:** Anirban Jana, Pierre Nassoy, Amaury Badon.

**Funding acquisition:** Anirban Jana, Kevin Alessandri, Gaëlle Recher, Pierre Nassoy, Amaury Badon.

**Investigation:** Anirban Jana, Naveen Mekhileri, Adeline Boyreau, Nadège Pujol, Amaury Badon.

**Methodology:** Anirban Jana, Naveen Mekhileri, Adeline Boyreau, Nadège Pujol, Gaëlle Recher, Pierre Nassoy, Amaury Badon.

**Project administration:** Kevin Alessandri, Pierre Nassoy, Amaury Badon.

**Resources:** Anirban Jana, Pierre Nassoy, Amaury Badon.

**Software:** Anirban Jana, Aymerick Bazin, Amaury Badon.

**Supervision:** Kevin Alessandri, Pierre Nassoy, Amaury Badon.

**Validation:** Anirban Jana, Gaëlle Recher, Pierre Nassoy, Amaury Badon.

**Visualization:** Anirban Jana, Gaëlle Recher, Amaury Badon.

**Writing – original draft:** Anirban Jana, Gaëlle Recher, Pierre Nassoy, Amaury Badon.

**Writing – review & editing:** Anirban Jana, Gaëlle Recher, Pierre Nassoy, Amaury Badon.

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
