## [Decision Letter · Decision Letter 0]

10 Jul 2024

PONE-D-24-10924zIncubascope: long-term quantitative imaging of multi-cellular assemblies inside an incubatorPLOS ONE

Dear Dr. Badon,

Thank you for submitting your manuscript to PLOS ONE. After careful consideration, we feel that it has merit but does not fully meet PLOS ONE’s publication criteria as it currently stands. Therefore, we invite you to submit a revised version of the manuscript that addresses the points raised during the review process.

We look forward to receiving your revised manuscript.

Kind regards,

Florian Rehfeldt

Academic Editor

PLOS ONE

Journal Requirements:

"This project was supported by grants from the French National Agency for Research (ANR-22-CE42-0019, ANR-21-CE18-0038, ANR-21-CE19-0029) and the Institut National du Cancer ( PLBIO 20-135 ). The authors also acknowledge the financial support from the Grand Research Program LIGHT Idex University of Bordeaux, the Graduate program EUR Light S & T PIA3 ANR-17-EURE-0027 and GdR ImaBio."

"We thank all the other members of the BiOf (Bioimaging and Optofluidic) team, in particular Camille Douillet, Aur´elien Richard, Laetitia Andrique, L´eon Rembotte, Fernanda Lopez-Garcia, for beta testing the device and suggesting improvements. We are also grateful to all the colleagues that encouraged us to develop a user-friendly 

version of the zIncubascope by expressing enthusiastically their need for such an instrument. This project was supported by grants from the French National Agency for Research (ANR-22-CE42-0019, ANR-21-CE18-0038, ANR-21-CE19-0029) and the Institut National du Cancer ( PLBIO 20-135 ). The authors also acknowledge the

financial support from the Grand Research Program LIGHT Idex University of Bordeaux, the Graduate program EUR Light S & T PIA3 ANR-17-EURE-0027 and GdR ImaBio."

"This project was supported by grants from the French National Agency for Research (ANR-22-CE42-0019, ANR-21-CE18-0038, ANR-21-CE19-0029) and the Institut National du Cancer ( PLBIO 20-135 ). The authors also acknowledge the financial support from the Grand Research Program LIGHT Idex University of Bordeaux, the Graduate program EUR Light S & T PIA3 ANR-17-EURE-0027 and GdR ImaBio."

4. Please upload a copy of Supporting Information Figure/Table/etc. S1-S5 Video which you refer to in your text on page 14/17.

Reviewers' comments:

Reviewer's Responses to Questions

**Comments to the Author**

1. Is the manuscript technically sound, and do the data support the conclusions?

Reviewer #1: Yes

2. Has the statistical analysis been performed appropriately and rigorously? 

Reviewer #1: Yes

3. Have the authors made all data underlying the findings in their manuscript fully available?

Reviewer #1: Yes

4. Is the manuscript presented in an intelligible fashion and written in standard English?

Reviewer #1: Yes

5. Review Comments to the Author

Reviewer #1: The manuscript presents an open-source and low-budget compact bright-field microscope built with off-the-shelf parts, which easily fits in a standard cell culture incubator and is suitable for several days-long time lapse recording of live specimens. The authors describe the setup and subsequently showcase applications with encapsulated cells, cysts, and yeast.

The element of novelty appears to be not really high, as the authors already published a very similar setup. The major improvement in this work is the integration of a motorized z-axis to record image stacks in bright field. I would suggest to consider the publication of the manuscript upon the accomplishment of major changes.

The positive aspects of this work are:

- Compact system built with off-the-shelf parts, it could be assembled and tested in few days provided that all the needed information for part purchase and assembly is given (which is not the case in the present version).

- Apparently stable over time and able to record sufficiently long time-lapse data in bright-field.

- Any open-source, low-cost solution to perform basic time-lapse live imaging experiments with bright-field illumination is positive fort he scientific community, especially for low-budget laboratories.

Criticisms/recommendations

- With one microscope only one condition can be examined. This limits its applicability to specific cases, so it’s not as universal as claimed and the authors should discuss it more critically.

- A part list for the building of the microscope is missing. A person skilled in opto-mechanics and microscopy will be able to identify the off-the-shelf parts immediately. However, users with less experience in these fields, who are an important target of the paper, will find very useful a list of parts with catalogue number and price, and the authors should add it.

- The GitHub repository contains the code, CAD data, and experimental data (e.g., time-lapse data). However, there is no explanation whatsoever, tips and tricks, troubleshooting, etc. Inserting these details in the repository will make the adoption of the system, particularly by inexperienced users, much easier. I suggest to revise the repository adding a comprehensive explanation to accompany the material. Without this, the repository is not useful.

- It would be recommendable to give hints on how to upgrade the system, e.g., with a motorized stage. In fact, a motorized xy-stage would be important for the community of users, and make the device more „universal “. The authors may want to comment on this.

- The system seems to be quite incremental with respect to the previous publication by the same authors. All the professional objective lenses have the same parfocal length and tube lens distance, so the previous system could already be used with other objective lenses, I suppose. The authors should clarify this point.

6. PLOS authors have the option to publish the peer review history of their article (what does this mean?). If published, this will include your full peer review and any attached files.

Reviewer #1: No

---

## [Author Response · Author response to Decision Letter 0]

19 Jul 2024

See the attached file that contains our responses to the editor and reviewer.

---

## [Decision Letter · Decision Letter 1]

5 Aug 2024

zIncubascope: long-term quantitative imaging of multi-cellular assemblies inside an incubator

PONE-D-24-10924R1

Dear Dr. Badon,

We’re pleased to inform you that your manuscript has been judged scientifically suitable for publication and will be formally accepted for publication once it meets all outstanding technical requirements.

Kind regards,

Florian Rehfeldt

Academic Editor

PLOS ONE

Additional Editor Comments (optional):

Reviewers' comments:

Reviewer's Responses to Questions

**Comments to the Author**

1. If the authors have adequately addressed your comments raised in a previous round of review and you feel that this manuscript is now acceptable for publication, you may indicate that here to bypass the “Comments to the Author” section, enter your conflict of interest statement in the “Confidential to Editor” section, and submit your "Accept" recommendation.

Reviewer #1: All comments have been addressed

2. Is the manuscript technically sound, and do the data support the conclusions?

Reviewer #1: Yes

3. Has the statistical analysis been performed appropriately and rigorously? 

Reviewer #1: Yes

4. Have the authors made all data underlying the findings in their manuscript fully available?

Reviewer #1: Yes

5. Is the manuscript presented in an intelligible fashion and written in standard English?

Reviewer #1: Yes

6. Review Comments to the Author

Reviewer #1: (No Response)

7. PLOS authors have the option to publish the peer review history of their article (what does this mean?). If published, this will include your full peer review and any attached files.

Reviewer #1: No

---

## [Editor Report · Acceptance letter]

20 Sep 2024

PONE-D-24-10924R1 

PLOS ONE

Dear Dr. Badon, 

I'm pleased to inform you that your manuscript has been deemed suitable for publication in PLOS ONE. Congratulations! Your manuscript is now being handed over to our production team.

Kind regards, 

on behalf of

Dr. Florian Rehfeldt 

Academic Editor

PLOS ONE